# Spermidine Synthase Localization in Retinal Layers: Early Age Changes

**DOI:** 10.3390/ijms25126458

**Published:** 2024-06-12

**Authors:** Astrid Zayas-Santiago, Christian J. Malpica-Nieves, David S. Ríos, Amanda Díaz-García, Paola N. Vázquez, José M. Santiago, David E. Rivera-Aponte, Rüdiger W. Veh, Miguel Méndez-González, Misty Eaton, Serguei N. Skatchkov

**Affiliations:** 1Department of Pathology and Laboratory Medicine, Universidad Central del Caribe, Bayamón, PR 00956, USA; astrid.zayas@gmail.com; 2School of Chiropractic, Universidad Central del Caribe, Bayamón, PR 00956, USA; 3College of Science and Health Professions, Universidad Central de Bayamón, Bayamón, PR 00960, USA; riosotero.ds@gmail.com; 4Department of Biochemistry, Universidad Central del Caribe, Bayamón, PR 00956, USA; 417adiaz@uccaribe.edu (A.D.-G.); david.rivera@uccaribe.edu (D.E.R.-A.); misty.eaton@uccaribe.edu (M.E.); 5Department of Natural Sciences, University of Puerto Rico-Carolina, Carolina, PR 00984, USA; 415cmalpica@uccaribe.edu (P.N.V.); jose.santiago13@upr.edu (J.M.S.); 6Charité–Universitätsmedizin Berlin, Institut für Zell- und Neurobiologie, Centrum 2, Charitéplatz 1, D-10117 Berlin, Germany; ruediger.veh@charite.de; 7Department of Natural Sciences, University of Puerto Rico, Aguadilla, PR 00603, USA; miguel.mendez3@upr.edu; 8Department of Physiology, Universidad Central del Caribe, Bayamón, PR 00956, USA

**Keywords:** polyamines, spermidine, spermidine synthase, nervous system, retina, glial cells, neurons, glial cell compartments

## Abstract

Polyamine (PA) spermidine (SPD) plays a crucial role in aging. Since SPD accumulates in glial cells, particularly in Müller retinal cells (MCs), the expression of the SPD-synthesizing enzyme spermidine synthase (SpdS) in Müller glia and age-dependent SpdS activity are not known. We used immunocytochemistry, Western blot (WB), and image analysis on rat retinae at postnatal days 3, 21, and 120. The anti-glutamine synthetase (GS) antibody was used to identify glial cells. In the neonatal retina (postnatal day 3 (P3)), SpdS was expressed in almost all progenitor cells in the neuroblast. However, by day 21 (P21), the SpdS label was pronouncedly expressed in multiple neurons, while GS labels were observed only in radial Müller glial cells. During early cell adulthood, at postnatal day 120 (P120), SpdS was observed solely in ganglion cells and a few other neurons. Western blot and semi-quantitative analyses of SpdS labeling showed a dramatic decrease in SpdS at P21 and P120 compared to P3. In conclusion, the redistribution of SpdS with aging indicates that SPD is first synthesized in all progenitor cells and then later in neurons, but not in glia. However, MCs take up and accumulate SPD, regardless of the age-associated decrease in SPD synthesis in neurons.

## 1. Introduction

The polyamines (PAs) spermidine (SPD) and spermine (SPM) are extremely important biomolecules and their content is critical for central nervous system (CNS) function [1,2,3,4,5,6]. PA levels decline during aging in some organs [7,8] while, in the CNS, it is relatively stable [7,9]. It is not clear if adult glial cells perform their own synthesis of PAs, or if these cells only uptake PA [10,11,12] via (i) low-affinity polyspecific organic cation transporters (OCT) such as SLC22A-1,2,3 [13] and SLC18B1 [14] and (ii) a high-affinity transporter ATP13A2/PARK9 [15,16], together with (iii) Cx43 hemichannels as an alternative pathway for PAs [17]. In fact, glial cells accumulate PAs [2,5,6,10,11,12]. Intriguingly, radiolabeled SPM [18] or biotinylated SPM (b-SPM) [6] cannot essentially pass through the blood–brain barrier (BBB). The limitations are probably due to (i) low-affinity transport and (ii) low concentrations of SPM and SPD (in the sub-micromolar range [19]) in the blood plasm. Indeed, the equilibrium constants (K50) of SLC transporters are a thousand times higher (mM range [13]) than the SPD plasm level. Conversely, b-SPM enters the brain from the cerebral-spinal fluid, and the astrocytes accumulate b-SPM [6]. This demonstrates an alternative PA pathway from external sources to the CNS. Natively, Müller retinal cells accumulate SPD/SPM [12,20], similar to astrocytes [21], and SPD dietary supplements help in guarding against cell malfunction in the aging CNS [22] and in retinal neuron survival [23].

In the retina, cone photoreceptor maturation [24] requires PAs, while the loss of the ATP13A2 transporter function in astrocytes contributes to neuronal pathology [25]. PA cations bind to negatively charged phosphates, DNA, RNA, and acid proteins [26,27], but retinal glial cells accumulate SPD/SPM to high levels [28] reaching an almost 1 mM concentration of free cations SPM+4 in adult Müller retinal glial cells [29]. In contrast, mechanical brain trauma [30], ischemia [31], and gliotoxins [32] cause the massive release of PAs from glial cells [32]. Catabolic PA degradation [7,31,32,33,34,35,36], oxidation, and the production of toxic acrolein [36,37] can lead to multiple CNS diseases such as Parkinson’s [38,39,40], Alzheimer’s, [41,42], and Huntington’s [43], and PAs are also key players in EAST/SeSAME, Snyder–Robinson, Down, and Rett syndromes [11,44,45]. However, unmodified PAs (i) protect against oxidative stress [46,47,48], (ii) glutamate toxicity [33], and (iii) DNA damage [36,49], and serve in (iv) improving memory [50,51] and (v) increasing longevity [52,53,54,55,56].

In the retina, PAs regulate the glial Kir4.1 channels in Müller cells [28,45,57,58,59,60] and can permeate Kir4.1 [59]. PAs open glial Cx43 channels [61,62,63] that are expressed in Müller retinal glial cells [63] and rescue Cx43 channels from hydrogen and calcium block. In addition, PAs may permeate Cx43 [61,62,63,64,65] and open a traffic pathway for other molecules [61,62,63,64,65]. Therefore, Müller cells, astrocytes, and the glial–neuronal communication in the retina need PAs [45,62,63]. PAs are extremely effective in various CNS functions [4,50,56,66,67], and SPD has been found to ameliorate or suppress oxidative stress in an animal model of optic nerve multiple sclerosis [68]. Inhibitors of PA oxidases, such as aminoguanidine, (i) help to restore facial nerves after injury in rats [69], while (ii) MDL72527 (spermine oxidase (SMOX) inhibitor) reduces DNA damage [36]. Restoring PA content in the retina and brain helps in trauma and aging [23,50,56,70,71]. Essentially, a daily intake of SPD reduces retinal ganglion cell death and enhances optic nerve regeneration following optic nerve injury [23], whereas PA deficiency disrupts retinal pigment epithelial cell migration and survival [72].

Intriguingly, retinal concentrations of SPD and SPM increase during development, followed by a decrease after 16 postnatal days [24,73]. Therefore, the purpose of this study was to (i) determine the localization of the SPD synthesis and (ii) examine possible changes in SPD synthesis with age in the rat retina. These are critical, since polyamine content in the CNS, specifically SPD, can promote longevity [53,54,55,56].

Previously, we showed that SpdS is localized in adult ganglion retinal cells [11] and in the synapses of adult brain neurons that are tightly surrounded by astrocytes [74,75]. However, there is a lack of data on (i) the age-dependent crosslink between storage and (ii) the synthesis of PAs in the retina [12]. Therefore, we investigated the localization and translocation of SPD synthase (SpdS) in the retina and SpdS age-associated changes. Since SPD controls many neuronal and glial receptors, age-dependent changes in the synthesis and content of PAs can be considered key elements of retinal function.

## 2. Results

### 2.1. Immunocytochemistry and Western Blot Analysis for Spermidine Synthase and Localization of Markers: Glutamine Synthetase in Glia, and DAPI in Nuclei

The retinae from rats in the neonatal period (3 days postnatal: P3), in the weanling period (21days postnatal: P21), and in early adulthood (120 days postnatal: P120) were analyzed for the expression of the biosynthetic enzyme spermidine synthase (SpdS). We used glutamine synthetase (GS), a specific glial marker, to identify the co-localization of spermidine synthetase in Müller retinal glial cells. DAPI, a marker of cell nuclei, was used to differentiate between retinal layers.

At the early postnatal age (P3) where cells are not yet matured, the staining process for DAPI (blue color, Figure 1a, left panel) shows two main cell layers: the ganglion cell layer (GCL, with newborn ganglion cells) and a neuroblast layer (NBL, the nuclei of most future neurons such as bipolar cells, horizontal cells (future inner nuclear layer (INL), photoreceptor cells, rods and cones (the future outer nuclear layer (ONL)). No DAPI staining was observed in the inner plexiform layer (IPL) because it is a future large synaptic zone. The outer nuclear layer (i) is not yet separated from the inner nuclear layer in NBL and (ii) the outer plexiform layer (OPL) has not yet formed.

Spermidine synthase, SpdS (the green color in Figure 1a), is already strongly expressed in (i) the area of newborn ganglion cells and (ii) in the neuroblast. At this point, young neurons are growing and differentiating from neuroglial progenitors in the NBL. SpdS is strongly expressed throughout the retinal tissue, specifically in the ganglion cell layer, where no overlap with the glial marker GS is observed. Inside the NBL, the nuclei of new neurons that synthesize spermidine are migrating along the glial fibers (Figure 1a; yellow arrows).

During this early postnatal stage, P3, we found the robust expression of glutamine synthetase, GS, (red labeling, Figure 1a) across all retinal layers. Glutamine synthetase is a vital glial marker, representing active glial radial cells. We did not use GFAP, another glial marker, because it represents reactive glia but not resting glia, while GS is a glial marker producing glutamine, which is released from glial cells and is used by neurons to produce the neurotransmitter glutamate. In P3 rats, GS was located across all layers (Figure 1a, red), given that most cells at this age are glial progenitor cells. It is also clear that glial cells surround all cell nuclei in the NBL.

When merged, the SpdS expression is mildly co-localized with GS (Figure 1a, right panel), only in the NBL and distal processes of Müller cells, but not in ganglion cells (GCL). At this stage, retinal cells in the NBL are mostly characterized by progenitors, which are not yet well differentiated into neuronal or glial types. Studying the exact type of neuronal cells is not the goal of the current study.

In P21 rats (Figure 1b) and P120 rats (Figure 1c), GS immunoreactivity was not observed in the neurons but was clearly present in all Müller cell compartments, including the somata, stalks, and distal processes in the outer nuclear layer (ONL), with strong expression in the INL (inner nuclear layer, where the bipolar cell and Müller cell somata are located), in the OPL (outer plexiform layer, where the synapses of photoreceptors are located) and in the ONL (outer nuclear layer, the layer of inner segments of photoreceptor cells, cones, and rods). Müller cells span the entire ONL, up to the outer segments of rods and cones (Figure 1b,c).

In P120 rats (Figure 1c), SpdS is most strongly expressed in the ganglion cells and their terminals and synapses in the IPL (Figure 1c, green). Weak labeling is observed in neurons of the INL. Several neurons in the INL have no SpdS staining. When merged, SpdS expression is not co-localized with GS (Figure 1c, right panel). At this stage, the ganglion cells are undergoing maturation. GS immunoreactivity is not observed in neurons but appears robustly in all Müller cell compartments, including the somata, proximal processes (stalks), and distal processes, with strong expression through the INL (where the Müller cell somata are located) and through the ONL (the layer of inner segments of the photoreceptor cells, cones, and rods). Müller cells extend their processes across all retinal layers up to the outer segments of rods and cones. GS labeling in P120 rats relates to mature glial radial cells.

As shown previously [12], in P120 rats, the accumulation of polyamine spermidine was found mostly in the proximal stalks of Müller cells around the ganglion cell body and in the endfeet of Müller cells, which also form the inner limiting membrane area (ILM) [12,20,29]. The ILM represents the Müller cell endfeet contacting the vitreous space of the retina where GS labeling is most intensive (Figure 1c), but no SpdS is found (Figure 1c, green). At this time, spermidine accumulation is found in precisely these compartments of Müller cells, where the glia lack SpdS, thus accumulating spermidine without synthesis. Intriguingly, in P120 rats, there were practically no spermidine labels in the ONL (the inner segment area of the photoreceptors) [12].

### 2.2. Western Blot Analysis for SpdS

Western blot analysis was used to quantify the changes in the concentration of SpdS within the whole of the retinal tissue at the three ages studied. Results showed that the amount of SpdS in the whole retina decreases with age (Figure 2). Retinae from P3 rats have 50% more SpdS protein present than the retinae of P21 and P120 rats. During early development from 3 to 21-days, SpdS activity drops dramatically, while, during the extended period of early adulthood from 21 to 120 days, the SpdS activity remains stable. There was no statistical difference in the percent fold measure between P21 and P120 rats. To evaluate the distribution of SpdS in each retinal layer, we used further semi-quantitative image analysis (Figure 3).

The loading control, β-actin, was used to normalize the data and justify stable conditions for all measurements of SpdS concentrations.

### 2.3. Analysis of SpdS Redistribution during Aging and between Retinal Layers

Images with SpdS fluorescence and co-localized images with both GS and SpdS obtained from confocal microscopy were analyzed using ImageJ software (version 2.1.0/1.53c), which allows for the comparison of signals. We measured the fluorescence spots in each area from three different regions of the retina where the most changes in SpdS labeling were observed during aging. For the SpdS images, we measured 15–20 fluorescence spots in 3 different cell types and regions of the retina: (i) ganglion cells in the end foot area; (ii) the inner nuclear layer (INL); and (iii) distal processes in the outer nuclear layer (ONL) in every image taken from the retinal samples. In every measurement, the images taken from retinal samples (*n* = 3) were analyzed (Figure 3a). It is clear that in P3 rats, the SpdS signal (Figure 3a, black columns) is slightly different between the GCL and distal processes (ONL), while in P21 (red columns), and specifically in P120 rats (grey columns), this difference became robust. For the co-localized images with GS and SpdS (Figure 3b), we measured 15–20 fluorescence spots in the inner segments of the photosensitive cells, rods, and cones located in the ONL. For both experiments, the fluorescence in each spot was measured and the mean of the spot fluorescence was imported to PRISM (Version 9.4.1 (458), GraphPad Software, San Diego, CA, USA) for statistical analysis. It is clear that there is practically no overlap of SpdS and GS signals at the P21 and P120 stages, while there is a strong overlap at the P3 age. The data clearly show the shift of SpdS labels from the multiple processes of progenitor cells and Müller cells at a young age to the neuronal body of ganglion cells, amacrine cells, and bipolar cells (in the GCL and INL retinal layers), but not in the distal processes.

## 3. Discussion

Recent advances in molecular biology have shown unique PA machinery and turnover in the CNS [6,17,21,35,76]. In this work, we studied SpdS behavior in the retina to find direct sources of spermidine production at different aging periods.

We made several key findings. (i) SpdS was expressed in glial progenitors during the early period of cell proliferation and differentiation (P3), but Müller cells showed no further synthesis of SPD during retinal maturation (P21) and, specifically, during yearly aging (P120). (ii) Since SpdS showed strong co-localization with neurons but not with the glial marker GS in Müller cells during maturation (P21) and during the early aging (P120) period, we concluded that neurons produce SPD. (iii) SpdS was almost exclusively expressed in ganglion cells (rarely occurring in other neurons) with age (Figure 1c), and such a translocation of SPD synthesis is associated with SPD accumulation in cell compartments called the endfeet of Müller cells, which surround the ganglion cells [12,20,28]. This indicates the local translocation of SPD from neurons to the glial endfeet. (iv) Most interestingly, the finding that, together with a near-complete decline with age of SPD content in neurons in the brain [74,75] and retina [12], the most important players that store SPD vitally are glial cells, regardless of a lack of synthesis in such adult glia.

In summary, such PA accumulation was observed in the Müller cells of different animals and in humans [11,12,20,28], showing stable evolution-associated SPD accumulation in such glia. Therefore, it is only at the earlier development stage that glial cells express PA synthesis. Similarly, cultured astrocytes showed the synthesis of polyamine putrescine (a precursor of spermidine) by ornithine decarboxylase (ODC); if ODC was inhibited, the astrocyte did not survive and did not proliferate. However, if PAs were restored by SPD supplementation [21], the survival and proliferation of astrocytes were restored. Such protection of growing glial cells by SPD was possible only if the transport systems were not inhibited from taking up SPD [21].

This makes Müller cells the key players in PA exchange. We suggest that if glial Müller cells hold spermidine regardless of the synthesis absence in such adult glia, such cells may be donors of SPD for neurons in critical times and may support neuroprotection when SPD is released from the glia. However, if glial cells lack such an accumulation (due to trauma or diseases), the aging neurons may not provide enough of their own SPD. Therefore, external sources of SPD are vitally important for retinal ganglion cell survival [23]. In the future, it will be necessary to investigate PA uptake/release mechanisms by glia using older stages of animal models and models of diseases.

Interestingly, the aging CNS suffers multiple disorders based on PA turnover malfunction, which is probably based on PA loss by glial cells. We, therefore, pose key questions for future investigation: (i) What are the mechanisms that underlie SPD redistribution and accumulation in different cells in the retina? (ii) What are the consequences of aging for SpdS relocation within the retina among the neurons and glia? Such a finding may help us to understand the dramatic PA-dependent neuronal network switch [77] and the roles of glial PA storage in retinal disorders and diseases [11,20]. Since the role of glia and SPD storage in retinal function and dysfunction are unclear, we demonstrated that there is no synthesis of SPD in the glia, while the storage of SPD was found in the glia [12]. Consequently, radial Müller glial cells are important during the development and neuronal migration of neurons, photoreceptors, and, possibly, other cells (astrocytes, endothelial cells, pigment epithelium, etc.), and are a main capacitor of SPD with aging in the retina [12]. As was observed in rabbit retinae, the concentration of PAs was found to decrease with age [24], and similar evidence was found in fractioned rat retinae [73].

In addition, glial Cx43 channels are facilitated for opening by PUT, SPD, and SPM [61,62,63,64,65], while glia-to-glia and glial-to-neuron communications are supported by PAs via connexin gap junctions between the retinal Müller glial cells [63] and between Müller cells and cones [63]. Since PAs regulate numerous ion channels [11,13,17,20,28,44,45,59,60,62,64,78,79], a release of PAs from the glia [17] may regulate neuronal receptors and channels [2,11].

Among the polyamines, SPD is found to be highest in the human brain [80] and increases in the plasm in aging humans [81], while biotinylated SPM does not permeate from the blood to the brain [6]. This mystery of such a PA increase in human plasm was interpreted as a marker of Alzheimer’s disease, but it may have the opposite meaning: that SPD is imported into the brain from the digestive system as a neuroprotector. Indeed, the intake of dietary SPD (i) increases cortical mass [82], restores age-induced memory impairment [50], improves cognitive performance [83], and restores the immune response of old human B cells [8]. Among its other roles, SPD also acts as an endogenous free radical scavenger, inhibiting the action of reactive oxygen species [39], restoring age-related memory impairment [50,55], preventing synaptic aging [56,67], preserving telomere length [84], and providing cardioprotection [54]. It was found that SPD boosts autophagy to protect from aging [55,56,66], and, as noticed above, spermidine dietary supplementation induces the regeneration of ganglion cells in the retina [23] and increases longevity [52]. However, if biotinylated SPM does not permeate the blood–brain barrier [6], there could be other as-yet unknown pathways to accumulate SPD and other PAs in the brain from either external or internal sources [6].

## 4. Materials and Methods

### 4.1. Animals and Tissues

Al experiments were performed under IACUC rules and approval and were in accordance with NIH requirements and the ARVO statement for the use of animals in ophthalmic and vision research. All procedures were carried out in accordance with the National Institute of Health guidelines for the humane treatment of laboratory animals and with the approval of the Universidad Central del Caribe Institutional Animal Care and Use Committee, protocol # 018-2021-04-00.

Eyes were obtained from Sprague–Dawley rats (3, 21, and 120 days old) that were housed in a standard cage in a 12-h light–dark cycle room and had access to food (standard rat chow) and water freely. The rats were sacrificed and the eyes were rapidly enucleated and then processed for immunohistochemistry and Western blot analysis.

### 4.2. Immunohistochemistry

Following enucleation, the eyes were fixed with two different solutions in glass vials for 45 min. The first solution (for spermidine synthase (SpdS) labeling) consisted of 4% paraformaldehyde (Sigma-Aldrich, P6148, St. Louis, MO, USA) in a phosphate-buffered solution at 0.1M (PBS: NaCl 136.9 mM, KCl 2.7 mM, Na2HPO4 10.1 mM, KH2PO4 1.8 mM with pH 7.4), The eyes were punctured with a 25 G needle in the ora serrata and fixed for an additional 20 min with fresh fixative solution. The eyes were washed 3 times with PBS 0.1 M before extracting the retinae in a cold PBS solution using a stereoscopic microscope (Fisher Stereomaster, FW00-20-1613, Waltham, MA, USA). The fixed retinae were embedded in 4% agarose (Gibco BRL, 15510-019, Thermo Fisher, Waltham, MA, USA) in PBS at 0.1 M. A Leica VT 1000 S Vibratome (Leica, Wetzlar, Germany) was used to create 20 μm retinal sections.

Samples were moved to a 24-well plate and permeabilized for 20 min with 1% DMSO (MP Biomedicals, 02196055, Santa Ana, CA, USA) and 0.3% Triton X-100 (Sigma, T9284) in PBS 0.1 M solution. The permeabilization solution was removed and replaced with a blocking solution containing 2% bovine serum albumin (BSA; Sigma, A4503) and either 5% normal horse serum (Vector, S-2000, Burlingame, CA, USA) or 5% normal goat serum (Vector, S-1000), depending on the secondary antibody, with 1% DMSO (MP, 196055) and 0.3% Triton X-100 (Sigma, T9284) in PBS 0.1 M for 1 h. The blocking solution was replaced with a fresh solution containing primary antibodies and left shaking overnight at 4 °C.

The following primary antibodies were used for the detection of glutamine synthetase (GS) (mouse anti-glutamine synthetase (Millipore, Burlington, MA, USA, MABN1182, dilution 1:250)); spermidine synthase (SpdS) (unconjugated rabbit polyclonal anti-spermidine synthase (Abcam, ab111884, dilution 1:250)) and SPD (Abcam, anti-spermidine ab7318, 1/100, Cambridge, UK) were used to determine SPD localization. (Note: since the SPD antibodies have a weak cross-reaction with SPM (reported by the company), we will use the terminology SPD immunolabel. The antibody for GS and for SpdS had strong immunoreactivity without cross-reactivity. After retinal section incubation with the primary antibody, the primary antibody aliquot was removed and the permeabilization solution was used to wash the samples 3 times for 10 min while shaking. Then, green fluorescence anti-rabbit FITC (Vector, Fl-1000, 1:200) and red fluorescence anti-mouse Texas Red (Vector, Tl-2000, 1:200) were used while diluted in the permeabilization solution and were added to the samples. The 24-well plate was covered from light and incubated for 2 h at 4 °C while shaking. Samples were washed 3 times for 10 min with PBS 0.1 M and once with distilled water. The tissues were then mounted on slides, left to dry for 5 min, and Fluoroshield with DAPI (Sigma, F6057) or Hard Set Vectashield with nuclear stain DAPI (Vector, H-1500) was added before sealing with coverslips. Therefore, all retinal samples were triple-labeled: with GS, a specific marker of glial cells; with SpdS, a marker of spermidine synthesis; and with DAPI, a marker of nuclei and retinal layers.

### 4.3. Confocal Microscopy

Confocal images were acquired using an Olympus BX60 microscope (Olympus, Tokyo, Japan) outfitted with the Olympus FV1000 confocal laser-scanning system. Images were taken using 40× magnification. To ensure veracity, experiments were performed in triplicate. All images were taken using image processing with the Fluoview program, ImageJ (NIH), when all key sets of confocal parameters were the same for all captured images. Adobe Photoshop was used to create the figures.

### 4.4. Western Blot Analysis

The tissue homogenization buffer (pH 7.5) contained (in mM): 20 Tris–HCl, 150 NaCl, 1.0 EDTA, 1.0 EGTA, 1.0 phenylmethylsulfonyl fluoride (PMSF), 1% Triton X-100, and an additional mixture of protease inhibitors (leupeptin, bestatin, pepstatin, and aprotinin). Proteins were separated on a 10% polyacrylamide–SDS gel (20 µg to 80 µg of protein/lane). Electroblotting was performed using the trans-blot turbo transfer system (cat. #1704150, Bio-Rad, Hercules, CA, USA) onto a nitrocellulose membrane. The membrane was blocked with Odyssey Blocking Buffer (cat. #927-40,000, LI-COR Biosciences, Lincoln, NE, USA) for 1 h at room temperature. The primary antibodies used to immunodetect the proteins assessed in this study were: anti-rabbit SpdS antibody; 1:5000 (cat. #ab111884, Abcam, Cambridge Science Park, Cambridge, UK) in blocking solution. These were incubated overnight at 4 °C. For the secondary antibodies, we used goat anti-rabbit 1:25,000 (cat. #926-32211, LI-COR Biosciences, Lincoln, NE, USA) for 1 h at room temperature. Our loading control was β-actin; therefore, we used an anti-mouse β-actin antibody, 1:5000 (cat. #A5441, Sigma-Aldrich, St. Louis, MO, USA). After that, we used goat anti-mouse 1:25,000 (cat. #926-32210, LI-COR Biosciences, Lincoln, NE, USA) as the secondary antibody for 1 h at room temperature. Infrared signals from the membranes were detected using the LICOR Odyssey model CLx scanner (LI-COR Biosciences, Lincoln, NE, USA). The membranes were stained with India ink for total protein to quantify small differences in the sample loading. Final detection was performed with the enhanced chemiluminescence methodology (SuperSignal^®^ West Dura Extended Duration Substrate; Rockford, IL, USA), as described by the manufacturer, and the signal was quantified using a gel documentation system (ChemiDoc, Bio-Rad, Hercules, CA, USA). The image was obtained using the Image Lab software (Version 6.1, Bio-Rad, Hercules, CA, USA).

### 4.5. Semi-Quantitative Image Analysis of Fluorescent Staining

All images were analyzed using ImageJ software (version 2.1.0/1.53c). We measured 7 fluorescence spots in 3 different regions of the retina: (i) the endfoot area, (ii) the inner nuclear layer, and (iii) the distal processes of the Müller cells (outer limiting membrane area) in every image taken of the retinal samples (*n* = 3). Images with the SpdS fluorescence and co-localized images with (GS and SpdS) obtained from confocal microscopy were analyzed using ImageJ software (version 2.1.0/1.53c). For the SpdS images, we measured 15–20 fluorescence spots in 3 different cell types and regions of the retina: (i) ganglion cells in the end foot area, (ii) neuroblast in the inner nuclear layer, and (iii) distal processes in the outer nuclear layer in every image taken of the retinal samples. For the colocalized images with GS and SpdS, we measured 15–20 fluorescence spots in the outer segments—the photosensitive segments of the photoreceptors. For both experiments, the fluorescence in each spot was measured, and the mean of the spot fluorescence was imported to PRISM (Version 9.4.1 (458), GraphPad Software, San Diego, CA, USA) for statistical analysis.

### 4.6. Data Analysis and Statistics

A two-way ANOVA with multiple comparisons (Tukey’s multiple comparison test) was used to compare the mean fluorescence of the SpdS samples and a one-way ANOVA with multiple comparisons (Tukey’s multiple comparison test) was used to compare the mean fluorescence of the colocalized samples. Statistical difference was established to be represented by *p*-values of lower than 0.05, with a 95% confidence interval.

## Figures and Tables

**Figure 1 ijms-25-06458-f001:**
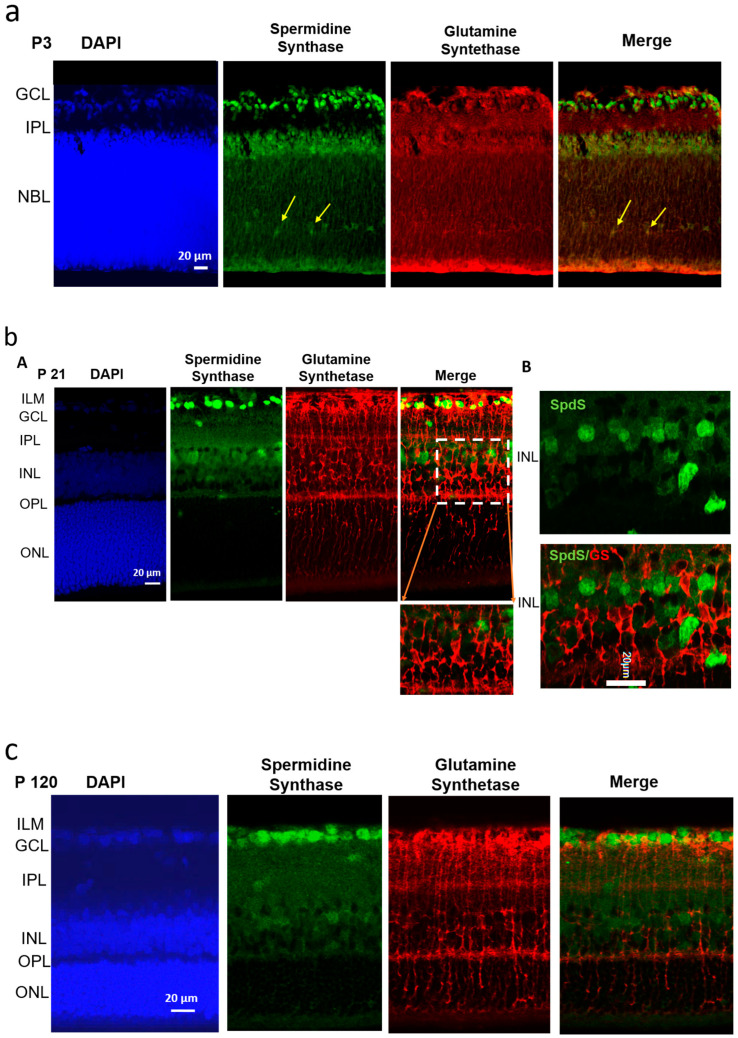
Localization of spermidine synthase (SpdS) in progenitors and adult neurons, but not in glial Müller cells. (**a**) Spermidine synthase (SpdS) and glutamine synthetase (GS) immunolabeling in a 3-day-old (newborn: **P3**) rat retina. SpdS (green) is strongly expressed in the nuclei of retinal progenitors, including new neurons migrating along the glial fibers (yellow arrows) in the inner plexiform layer (IPL) and in the neuroblast layer (NBL). SpdS is, therefore, expressed in all cell types at this early stage of retinal development. (Scale = 20 µm; GCL—ganglion cell layer; IPL—inner plexiform layer, the developing synaptic area; NBL—neuroblast layer, the proliferative zone of the inner optic cup, consisting of retinal progenitor cells). (**b**) Spermidine synthase (SpdS) and glutamine synthetase (GS) immunolabeling in a 21-day-old (weanling: **P21**) rat retina. (**A**)—SpdS (green) is localized in the neuronal cells: in ganglion cell bodies and in cells in the inner nuclear layer, while GS is in the glial cells. No co-localization was found between GS and SpdS. The insert shows Müller cells (red) and no co-localization with SpdS (green). (**B**)—The enlarged representation of the absence of co-localization of SpdS and GS, taken randomly from the inner nuclear layer of a P21 rat. (Scale = 20 µm; ILM—inner limiting membrane, where the endfeet of Müller cells make a border between the retina and vitreal humor; GCL—ganglion cell layer where the ganglion cell bodies are surrounded by Müller cell processes; INL—inner nuclear layer, the neuronal bodies, the cell nuclei of young bipolar, horizontal cells and Müller cells; OPL—outer plexiform layer, the synaptic area between photoreceptors and bipolar cells; ONL—outer nuclear layer, the bodies of rod and cone photoreceptors). (**c**) Spermidine synthase (SpdS) and glutamine synthetase (GS) immunolabeling in a 120-day-old (young adult: **P120**) rat retina. SpdS (green) has been localized in the neurons (ganglion cells, bipolar cells and synapses), but not in the Müller cell compartments. Compared to the P21 retina (**b**), SpdS is less pronounced in the neurons of these older rats. (Scale = 20 µm; ILM—inner limiting membrane where the endfeet of Müller cells make a border between the retina and vitreal humor; GCL—ganglion cell layer; INL—inner nuclear layer, the neuronal bodies of bipolar, horizontal cells and Müller cells; OPL—outer plexiform layer, the synaptic area between photoreceptors and bipolar cells; ONL—outer nuclear layer, the cell nuclei of the rod and cone photoreceptors).

**Figure 2 ijms-25-06458-f002:**
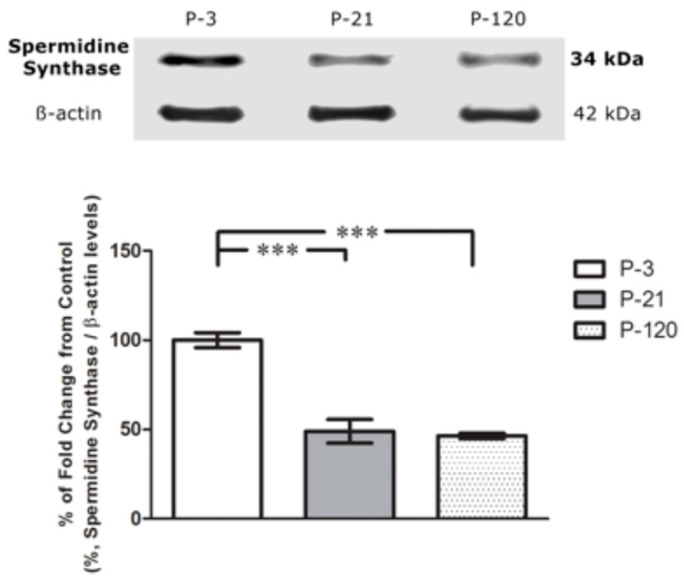
Western blot analysis of SpdS content during aging in the whole retina. Quantification of spermidine synthase (SpdS) using Western blot analysis. The graph displays the quantification of the chemiluminescence intensity ± the standard error of the mean (SEM) of SpdS protein in whole retina lysates from rats of 3, 21, and 120 days old. SpdS was detected as a band at 34 kDa, which is consistent with the predicted molecular weight of spermidine synthase. The results of 3 separate experiments using different rats are shown. The asterisks indicate a significant difference from control (P3) to (P21) and (P120), but no difference was seen between P21 and P120. (one-way ANOVA). β-Actin was used as a loading control to normalize the data. (One-way ANOVA results: ***; *p* < 0.001, values depicted are mean + SEM; *n* = 3, *p* = 0.0003; F (2,6) = 43.40).

**Figure 3 ijms-25-06458-f003:**
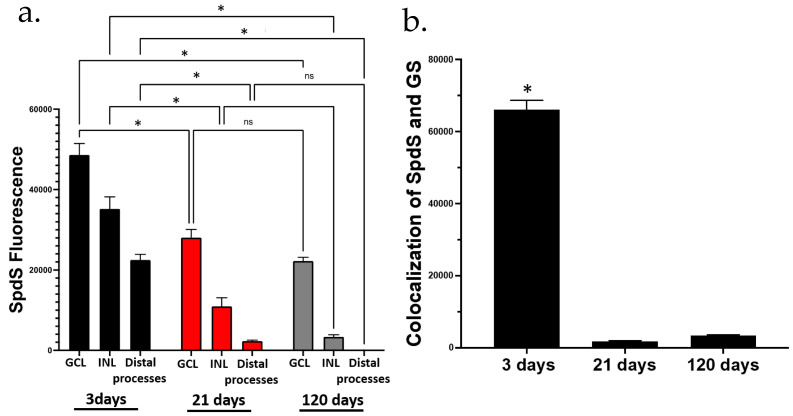
Semi-quantitative analysis of SpdS redistribution during aging and between retinal layers from 3-, 21-, and 120-day-old rat retinae. (**a**) SpdS fluorescence comparison between three retinal layers (ganglion cell layer (GCL), inner plexiform layer (IPL), and neuro blast (NBL). The NBL contains the future inner nuclear layer (INL) and outer nuclear layer (ONL) where the distal processes of Müller cells are passing through. In the graph, the statistical difference within a group (same day) is shown. In addition, there were statistical differences among the groups in the endfoot (P3 vs. P120 and P21 vs. P120) and somatic area (P3 vs. P21 and P21 vs. P120). Distal processes (ONL) did not present any statistical difference. (**b**) Colocalization of the fluorescence showed by SpdS and GS was measured. The 3-day-old rat retina demonstrated major colocalization compared with 21- and 120-day-old rats’ retinae. In both panels (**a**,**b**), the samples with an asterisk (*) indicate *p*-values lower than 0.05 with a 95% confidence interval, which was considered statistically different. (ns) indicates that there was no statistical difference among the samples.

## Data Availability

Additional data supporting the conclusions of this finding will be made available without undue reservation.

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
