# Peer review of "Spermidine Synthase Localization in Retinal Layers: Early Age Changes"

_ijms, 2024, doi:10.3390/ijms25126458_

Round 1

Reviewer 1 Report

Comments and Suggestions for Authors

The authors present work investigating the localization and expression of the SpdS enzyme in rat retinas at three time points. This straightforward experimental design reveals that SpdS is primarily expressed by retinal ganglion cells as well as yet-to-be identified cells within the inner nuclear layer. Unfortunately, what could have been a very elegant and simple to present work is muddled by far, far, far too much background and speculation. The authors must significantly shorten their review of the literature and reduce the amount of speculation on what their results mean within the broader context of the field.

Introduction

In general, the introduction is very confusing to read. The whole section needs significant editing for both English and content. In addition, there are 98 references included in the introduction alone. This is far in excess of what is required to establish the importance of PAs in aging, and especially in the retina. The authors need to revise this entire section to focus on PA biology in the retina specifically, and why changes in PA levels with aging is a problem for the retina.

It isn’t clear why Polyamines are being investigated at all. The authors state that their function is a mystery, but that their levels change with aging (except within the CNS). Later, the authors state that it is essential for eIF5A activity and for glial cell survival and proliferation. The authors should clearly state the importance of PAs. The authors further go on to list 11 known functions/roles of PAs in cellular biology in the third paragraph. The authors must rewrite the introduction and focus on the cellular roles of PAs that are important for the retina. This is written as a broad review of PAs in general.

The introduction should be written to briefly indicate why PAs are important. They must then go on to discuss what is known about their age-related expression in the CNS and/or other organs, and finally, what is known about its expression within the retina. This establishes a reason to do this experiment (that the expression of SpdS, and important enzyme in PA synthesis not known in the retina).

Results

Figure 1B – this sub-panel consist of two sub-sub panels. In addition, the IF images on Fig 1bB labels SpdS as SDS. The authors need to re-configure this figure so that there are only sub-panels. The figure legend is confusing to read and contains more information than the results section in the body of the paper. The authors should not be drawing conclusions in the figure legend.

Figure 1D is essentially new data of what the authors have published previously. It isn’t clear why it is included here. This sub-panel should be removed.

Sections 2.1, 2.2, and 2.3 should be combined into a single section and be written in order of time point and marker expression. The way it is written currently forces the reader to flip back and forth between sub-sections.

It is not clear that GS and SpdS co-localized in Figure 1A. The resolution/magnification is not high enough to conclude this. At the same time, the merged imaged strongly suggest that they do not co-localize, which is consistent with their later timepoints.

It would be very informative for the authors to stain with additional markers of inner-retinal neurons (horizontal cells, bipolar cells) to determine which sub-types are expressing SpdS.

The sentences at lines 224-228 are very confusing to read.

Figure 2 – Could the authors please provide the original western blot image(s) which were used for quantification?

Figure 3 – Semi-quantitative analysis of IF images is very difficult to do, and in the opinion of this reviewer, not informative without some way to normalize the intensity of the images. How did the authors normalize fluoresecent intensity between time points to determine a decrease? At best, this can be used to show differences in fluorescence between regions within the same timepoint. The western blot in figure 2 already shows a decrease in SpdS expression over time. The authors should not be comparing between time points in this figure.

Figure 3 – 21 and 120 days – there is no neuroblast layer at either of this time points. What layer are the authors referring to?

 Discussion

The first paragraph of the discussion belongs in the introduction. Interestingly, as all of these enzymes are required to synthesize SPD, is there any evidence that these enzymes are expressed in the retina (specifically retinal ganglion cells and the cells in the inner retina that express SpdS)?

The second paragraph of the discussion belongs in the introduction. With these two paragraphs in the introduction, that brings the total number of references to 107 just for the introduction. This is far, far too detailed for a relatively straightforward manuscript examining the expression of an enzyme at 3 different timepoints.

The third paragraph of the discussion belongs in the introduction. There still has been no discussion of the data presented in the manuscript.

The fourth paragraph of the discussion does not discuss their data, but could if it were slightly tweaked. Otherwise, it belongs in the introduction. Interestingly, the data presented by the authors here, and in their previous manuscript (PMID: 37189626) contradict the statement the SPD is protective for RGCs since they see accumulation of SPD in Mueller glia and not RGCs.

The sixth paragraph belongs in the introduction and does not discuss the data presented in this manuscript.

Paragraph 7 needs to be re-written. The authors only identified the distribution of SpdS in the rat retina. They did not show the effects of PA on channels and transporters, their role in CNS disorders, or the importance in the glial system. This is not a review article. This reviewer is not convinced that SpdS is ever expressed in glial cells from the data presented in this paper.

The discussion needs to be completely re-written to actually discuss the data presented in this manuscript within the context of SPD synthesis in the retina.

Methods

The authors must list which institution the IACUC is a part of, and what the protocol number is that these experiments were performed under.

Section 4.2 – The authors state that two separate fixative solutions were used, but only describe the fixative for staining for SpdS. I assume that the SPD staining requires fixation in a different fixative. Can the authors please add this to the methods section.

4.4 – Why is there such a large variation in the amount of protein loaded per well? There was only a single western image shown. Can the authors provide more details on the exact amount of protein loaded in each lane for each sample? Can the authors please also provide the india ink image that was used to determine small variations in protein content?

Conclusion

This manuscript should be significantly shortened to reflect the straightforward experiments performed by the authors. Once this is done, the conclusion section will no longer be needed and can be integrated into the discussion section.

Comments on the Quality of English Language

This manuscript contains many serious flaws in sentence structure, grammar, and spelling. It can be difficult to follow the authors' line of thought throughout. Significant work is needed to remedy flaws in logical flow of the introduction, results, and discussion.

Author Response

"Please see attachment"

Reviewer 2 Report

Comments and Suggestions for Authors

Authors described the localization of spdS and GS in the retinal layers of postnatal days 3, 21, and 120 of rat retinas. Some suggestions were as followings.

(1) The original images in the Figure 1 (a to d) of DAPI, SpdS, GS and merge should be the same size in order to compare with each other. The retinal cells and muller glia in the Fig 1d should be combined as Figure 1a and others were as the Figure 1b to 1e.

(2) The levels of Put, Spd, Spm in the retinal layers  of postnatal days 3, 21, and 120 should be provided to discuss the correlations with the SpdS and GS.

(3) The title ".....Aging Changes" is not clear.  If the data contain postnatal days 150 or 180, the term aging might be more suitable. The postnatal day 120 might be in the growth stage.

Round 2

Reviewer 2 Report

Comments and Suggestions for Authors

Acceptable in the present format.

Author Response

Thank you very much.